# Analysis of Geometric Surface Structure and Surface Layer Microhardness of Ti6Al4V Titanium Alloy after Vibratory Shot Peening

**DOI:** 10.3390/ma16216983

**Published:** 2023-10-31

**Authors:** Jakub Matuszak

**Affiliations:** Department of Production Engineering, Mechanical Engineering Faculty, Lublin University of Technology, Nadbystrzycka 38D, 20-618 Lublin, Poland; j.matuszak@pollub.pl; Tel.: +48-81-538-4707

**Keywords:** shot peening, geometric surface structure, surface layer, microhardness, topography, surface roughness

## Abstract

This article presents an analysis of the impact of vibratory shot peening on the surface roughness and physical properties of the Ti6Al4V titanium alloy surface layer after milling. The elements of machine parts and structures made of titanium alloys are often exposed to variable loads during operation. Therefore, it is advisable to apply methods that enhance functional properties and increase the durability of interacting components. Increasing the operational durability of such elements can be achieved by vibratory shot peening. Variable amplitudes A = 24; 33; 42; 51; 60 mm and times t = 1; 7; 13; 19; 25 min were applied. It has been demonstrated that it is possible to achieve a threefold reduction in the roughness parameter, Sa = 0.344 µm, compared with milling, Sa = 0.95 µm. An increase in Smr(c) areal material ratio was observed after vibratory shot peening compared with milling. It has been shown that amplitude has a greater impact on the increase in hardening of the surface layer g_h_ compared with time. The highest rate of change in surface roughness and thickness of the hardened layer was achieved at a vibratory shot-peening time of t = 13 min. The greatest thickness of the hardened layer, exceeding 200 µm, was obtained after shot peening with an amplitude of A = 60 mm.

## 1. Introduction

Titanium alloys, despite their significant cost, find a relatively broad range of applications. Properties such as high strength-to-weight ratio, maintaining good strength properties at elevated temperatures, and very high corrosion resistance make these materials well-suited for applications in the aerospace, maritime, chemical, automotive, and biomedical industries [1]. Titanium alloys are used to manufacture numerous components for aircraft engines, airplanes, and helicopters, with a significant portion of the world’s titanium production finding applications in the aerospace industry [2]. An important property of titanium alloys is their biocompatibility, which enables their use in the production of medical implants and surgical instruments [3,4,5]. Greater potential for application of titanium alloys in the biomedical industry can be achieved by modifying the chemical composition, replacing toxic components with biocompatible ones [6]. Aircraft components made of titanium alloys often have complex shapes and meet high requirements regarding accuracy and surface quality. To meet these requirements, the elements are shaped using machining methods [7]. Titanium alloys are classified as difficult-to-cut materials [8,9]. This is due to the properties of titanium such as low thermal conductivity, high chemical reactivity at elevated temperatures, and the tendency to form built-up edge during machining [10]. These properties lead to a rapid wear of cutting tools, resulting in their limited durability [11]. In the process of milling titanium alloys using carbide tools, the tool wear is significantly influenced by the grain size of these carbides [12]. Catastrophic tool wear is particularly dangerous in the cutting process. Understanding the mechanics of fracture in carbide cutting inserts can help prevent this phenomenon [13]. Titanium has a relatively low longitudinal elastic modulus, making it susceptible to deformation during machining. The need to reduce the weight of aircraft components involves the use of thin-walled components. To ensure the required accuracy and surface roughness of such elements, the use of appropriate machining conditions must be applied [14,15,16]. Due to a series of dynamic phenomena occurring in the process, as the thickness of the machined elements decreases, there is an increase in the roughness of the machined surface. A comparison of surface roughness after milling the thin-walled components made from different materials indicates that the surface roughness of components made from Ti6Al4V titanium alloy is lower than the surface roughness of components with the same thickness made from composite material with carbon fibers and EN-AW-2024 aluminum alloy [17].

Functional properties of machine components can be enhanced as a result of surface layer hardening. If hardening is obtained as a result of static impact of the tool on the workpiece, such treatment is called burnishing [18]. However, when hardening is caused by pressing elements (e.g., balls) that impact the processed object, the shot-peening process takes place [19]. Shot peening can be carried out in a regular or random manner [20]. Both burnishing and shot peening change the surface layer properties of processed objects [21,22]. Under the influence of burnishing tools or shot-peening elements, a new geometric structure of the surface is formed [23]. However, plastic deformations of the surface layer are associated with an increase in the microhardness of this layer and the formation of compressive residual stresses [24]. The burnishing and shot-peening processes are tested both by experimental methods and FEM numerical modeling techniques [25].

Titanium alloys are subject to various surface treatment methods [26,27]. The surface layer of titanium alloy elements is changed by both burnishing and shot-peening methods. According to Revenkar et al., ball burnishing with cemented carbide balls of titanium alloy Ti6Al4V resulted in an increase in the surface layer microhardness, as well as an improvement in wear resistance [28]. In the studies in [29,30], the beneficial effect of ball burnishing on the surface structure and properties of the surface layer of titanium alloys has been demonstrated. Titanium alloys were also subject to slide burnishing. Toboła et al. [31] presented the effects of using slide burnishing and low-temperature nitriding on the Ti6Al4V alloy, while Zaleski et al. [32] provided research results on the influence of slide burnishing with a surface-active liquid lubricant on the fatigue durability of the Ti6Al2Mo2Cr titanium alloy. The properties of the titanium alloy surface layer are often changed by shot blasting. Liu et al. [33] studied the effect of air pressure and process duration on the surface roughness of the Ti6Al4V alloy after shot blasting. Ongtrakulkij et al. [34] examined the condition of the surface layer after double shot peening. Studies on the influence of various processing methods on the titanium alloy TB6 have shown that the best results are achieved for samples subjected to sequential milling, polishing, shot peening, and polishing again [35]. Chen et al. found that wet shot peening with ceramic balls of the Ti6Al4V alloy allows obtaining lower surface roughness than dry shot peening [36]. The research conducted by Wang et al. showed that shot peening can improve the fatigue performance of the Ti6Al4V titanium alloy [37]. Zaleski demonstrated an increase in fatigue life as a result of vibratory shot peening [38]. Shi et al. found that the use of combined classic shot peening and vibratory shot peening of the TC17 titanium alloy allowed for a greater increase in fatigue strength than after classic shot peening alone [39]. The improvement of fatigue properties of the TA15 titanium alloy was achieved by shot peening and subsequent nitrogen ion implementation [40]. Aguado-Montero et al. investigated the influence of shot peening, laser peening, and chemically assisted surface enhancement on the surface roughness, residual stresses, hardness, and fatigue strength of the Ti6Al4V titanium alloy, produced using an additive manufacturing method [41]. Avcu et al. found that shot peening enhances the wear resistance during dry sliding friction of titanium alloy specimens manufactured via powder metallurgy [42]. Meanwhile, Yang et al., based on their research on the influence of shot peening at various intensities on the fretting wear of Ti6Al4V alloy, demonstrated that during the early stages of wear, shot peening increases the wear rate, while in the later stages, it reduces the wear rate [43]. In turn, Zhang et al. [44] presented the results of research on the impact of shot peening of titanium alloy Ti 811 on fretting fatigue life at elevated temperature. Research by Żebrowski et al. [45] showed that the technological conditions of shot peening of titanium alloy samples produced using the additive method have an impact on corrosion resistance.

The analysis of publications shows that due to the physical properties of titanium alloys, the final shaping of elements made of these alloys is mainly carried out by machining methods. Due to the variety of milling variants, machining strategies and tools that can shape complex geometric structures, milling predominates as the “first choice” method. Due to the physical properties of titanium alloys, they are used as critical elements in the aviation, automotive, and biomedical industries, with very high requirements regarding quality and durability. Therefore, it is advisable to use methods that enhance functional properties and increase the durability of mating components. The elements of machine parts and structures made of titanium alloys are often exposed to variable loads during operation. Increasing the operational durability of such elements can be achieved by shot peening. Improving the surface layer properties through shot peening can also enable the use of smaller cross-sections due to increased operational durability (caused by vibratory shot peening), which is highly significant in terms of reducing the weight of structures in the automotive and aviation industries. Due to the limited amount of research on the effects of vibratory shot peening after milling, this paper provides a detailed analysis of process parameters on the surface layer properties, thus offering the opportunity to adjust the parameters to generate the desired properties of elements made of Ti6Al4V titanium alloy.

## 2. Materials and Methods

A general diagram of the experiment is presented in Figure 1. The research focused on analyzing the impact of variable vibration peening conditions on surface roughness, as well as selected parameters of the surface layer hardening after the process.

The initial processing consisted in face milling. To remove the technological history, after preparing the samples to the required dimensions, annealing was carried out in the vacuum furnace RVFOQ-424 for 60 min (0/+5 min) at a temperature of 704 °C (+/−14 °C) at a pressure below 13 MPa. The initial cooling to 200 °C was conducted in a vacuum, and below 200 °C, the samples were cooled in the open air until they reached room temperature.

### 2.1. Tested Material

The tests were carried out on samples made of Ti6Al4V titanium alloy. The chemical composition and basic physical properties of the tested material are presented in Table 1, while the shape and dimensions of the samples are shown in Figure 2.

### 2.2. Surface Treatment before Shot Peening

One of the sample surfaces, with dimensions of 15 × 100 mm (later intended for shot peening), was subjected to face milling. The tests were conducted on the Avia VMC800HS three-axis machining center. For the milling process, the Fraisa 20 mm diameter four-blade solid carbide milling cutter (symbol: HM MG10 γ = 5°; λ = 40°) was used. The following cutting parameters were applied: v_c_ = 70 m/min, a_p_ = 0.7 mm, a_e_—15 mm, f_z_ = 0.05 mm/tooth. After milling, the sample dimensions were 4.3 × 15 × 100 mm. The process of milling the samples was carried out using new cutters, which were replaced with new ones after exceeding the index VB3 = 0.15 mm (VB3—flank wear according to ISO 8688-2: 1996 standard).

### 2.3. Shot-Peening Process

Vibratory shot peening was carried out on a special workstand, which is schematically shown in Figure 3.

The tested samples were mounted in the working chamber in such a way as to enable shot peening of the previously milled surfaces. The bearing steel balls with a hardness of 60 HRC and a diameter of 9 mm were used as the machining medium. The vibration frequency was constant during the experiment and was set at 7Hz. The variable parameters were the vibration amplitude and the shot-peening time (Table 2). The vibratory shot peening was carried out without additional machining fluids. The samples were mounted using two pressure strips located at opposite ends of the samples.

### 2.4. Methodology for Measuring the Surface Structure and Surface Layer Properties

The surface topography was measured using a T8000RC120-400 profilographometer provided by Hommel-Etamic Jenoptik (Jena, Villingen-Schwenningen, Germany). The surface topography was tested on the surface with dimensions of 4.8 mm × 4.8 mm using the contact method after vibratory shot peening with variable vibration amplitude and time. The roughness parameters, such as Sa (arithmetical mean height) and Smr(c) (areal material ratio), were measured. The areal material ratio Smr was obtained for c = 40% of Sz, which is demonstrated in Figure 4. Based on the measured values, the average and standard deviations of the obtained measurements were determined.

The surface microhardness was determined with the Leco LM700 device in compliance with the EN-ISO 6507-1:2018 standard. The Leco tester was equipped with a 10× magnification eyepiece and a 50× magnification lens. The penetrator loading time was 15 s with a 50 g load applied. Microhardness measurements were made on oblique sections, which allowed for a significant extension of the measurement surfaces compared with sections perpendicular to the machined surface. The surfaces were prepared on an SPD-30 grinder at an angle of 3° (Figure 5).

The following indicators were determined based on the microhardness distribution in the surface layer of the tested samples (Figure 6):-Microhardness increase ΔHV0.05;-Thickness of the hardened layer g_h_.

**Figure 6 materials-16-06983-f006:**
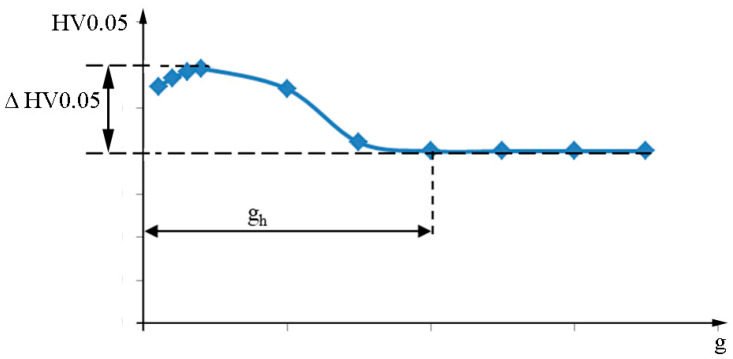
Indicators characterizing the distribution of microhardness in the surface layer of tested samples.

A microhardness increase of ΔHV0.05 is the value of microhardness that increased in relation to the core of the material caused by the vibratory shot-peening process. The majority of damages to machine components begin at or directly below the surface. The increase in microhardness and the depth of the hardened layer are indicators of the surface’s resistance to this type of damage.

## 3. Results and Analysis

### 3.1. Surface Topography

Figure 7a,b shows the surface topography after milling and vibratory shot peening, respectively. The distribution of micro-irregularities on the milled surface results from the geometry of the blade and its diameter, as well as the machining kinematics and, in particular, the feed per tooth. The average value of roughness parameters after milling was Sa = 0.95 µm and Sz = 8.9 µm.

The surface after vibratory shot peening has a nondirectional structure obtained as a result of the balls hitting the surface (milling marks are invisible). The pockets left by the impacting balls allow the formation of microlubrication grooves for the mating elements.

Figure 8 shows surface roughness after vibratory shot peening. As the amplitude increased (in the initial phase, up to A = 33 mm), the surface roughness decreased. This is due to the fact that as the amplitude increases, the speed and thus the kinetic energy of the balls hitting the surface increases. Greater energy allows for a more effective removal of micro-irregularities created after the milling process. The location of the minimum roughness area is correlated with the roughness after the previous machining. It can be expected that the lower the roughness after the preceding machining process, the more minimum the roughness value achieved for smaller amplitudes will be during vibratory shot peening. If the surface roughness peaks are larger after the preceding machining process, greater kinetic energy (thus, a larger amplitude in the peening process) will be required to effectively improve the surface quality.

With an increase in time (Figure 8b) up to a value of 13 min, there was a decrease in surface roughness. With the increase in vibratory shot-peening duration, the number of impacts per unit of surface area also increases. However, for longer peening durations, a slowdown in the decreasing trend of roughness was observed. This is due to the “saturation” of the surface being impacted by the striking balls, and further increase does not result in geometric structure changes. For this reason, for the machining parameters applied, the most optimal time for roughness improvement is t = 13 min. A similar trend was observed for the Sz parameter (Figure 9). It is important to note that within the entire range of the parameters applied, lower roughness parameter values were achieved compared with milling.

In Figure 10, a comparison of the areal material ratio curve after milling and vibratory shot peening is presented: for c1 = 0.4*Sz after milling Smr (c1) = 43.8% and Smr(c1) = 63.3% after shot-peening process.

The sharp peaks on the surface caused by the replication of cutting tool edges during milling influenced the lower Smr value after milling.

Figure 11 shows the influence of processing conditions on the areal material ratio. Throughout the entire range of the vibratory shot-peening parameters applied, there was an increase in the Smr ratio compared with the milling process. 

This means that in terms of mating and wearing surfaces, the contact area is larger, and consequently, the surface is more resistant to wear. Additionally, a characteristic feature of the shot-peening process is the creation of microgrooves that retain oil droplets, which can further contribute to extending the durability of mating components. A greater slope of the curve in the core area after the milling process implies a faster rate of wear for such a surface. 

### 3.2. Surface Layer Microhardness

Figure 12 shows the comparison of microhardness distribution of the surface layer after milling and vibratory shot peening. Changes in hardness in the near-surface zone of a material after the milling process depend on the phenomena occurring during the separation of the cut layer and its transformation into a chip [46]. These phenomena include machining parameters and tool geometry and wear, as well as the use of machining fluid. The blade geometry (rake angle, corner radius, and edge radius) influences the distribution of cutting force that affects the surface layer. 

After milling, a characteristic increase in microhardness of the surface layer just below the surface (at a depth of 20–30 μm) was observed. The mechanical impact of the cutting blades on the workpiece material contributes to the plastic deformation of material and increases temperature in the area of contact between the tool and the workpiece, which always affects surface layer properties. In turn, due to the specific nature of the vibratory shot-peening process, the impact of the striking balls on microhardness and the depth of the hardened layer is significantly greater compared with milling.

Figure 13 shows the microhardness increase after vibratory shot peening. The speed of the balls moving in the chamber increases with an increase in amplitude (Figure 13a). This results in an increase in the kinetic energy of the peening elements that hit the sample surfaces. The precise determination of energy is difficult due to the process randomness. The freely moving balls lose energy by colliding with each other or with the walls of the chamber, thus changing the direction of their movement and speed. In turn, as the vibratory shot-peening time increases (Figure 13b), the number of impacts per sample surface increases too, i.e., the degree of ball impact coverage on the surface also contributes to the hardening of the surface layer. In both cases (influence of amplitude Figure 13a,b), the microhardness significantly exceeds the values obtained after the milling process.

Figure 14 shows the thickness of the hardened layer g_h_ after vibratory shot peening. The specific properties, purpose, and effect of using the shot peening and burnishing processes result in a hardened layer thickness that significantly exceeds the thickness after milling. Figure 14a,b shows the influence of the vibratory shot-peening amplitude and time on the thickness of the hardened layer g_h_, respectively. The g_h_ parameter varies from about 80 (for an amplitude A = 24 mm) to over 200 μm (for A = 60 mm).

The influence of shot-peening time on the thickness of the hardened layer is also visible (Figure 14b). The largest increase is observed up to the 13th minute of the process, and a further increase in time does not significantly affect the increase in the hardened layer thickness. Amplitude has a greater impact on the increase in hardened layer thickness compared with shot-peening time.

In the range of this research conducted on the influence of input factors (amplitude and time) on outputs, it can be observed that a change in the amplitude of vibratory shot peening significantly affects the parameters of roughness, microhardness increase (ΔHV0.05), and the thickness of the hardened layer g_h_ throughout the entire range of amplitude values adopted in this experiment. On the other hand, the process time introduces the most significant changes in effects during the initial phase of treatment, and further increasing the time of vibratory shot peening does not bring significant changes. Considering the efficiency and process optimization, a favorable impact on the technological surface layer is achieved up to t = 13 min.

## 4. Conclusions

This article presents an analysis of the impact of vibratory shot peening on the surface roughness and physical properties of the surface layer of Ti6Al4V titanium alloy after milling. The tests were carried out in a range of different amplitudes affecting the kinetic energy of the impact and time. The following results summarize this study of the effect of vibratory shot peening on the surface roughness and microhardness of Ti6Al4V titanium alloy:Vibratory shot peening leads to a reduction in roughness in the entire range of process conditions used in the experiment;For the value A = 33 mm and time t = 13 min, the lowest roughness value was obtained (Sa = 0.344 µm); this represents approximately a threefold reduction in the parameter compared with milling;The effects after the vibratory shot-peening process are correlated with the roughness after the previous machining;An increase in amplitude above A = 33 mm (and, consequently, the kinetic energy of impact) leads to an increase in surface roughness, which is caused by plastic deformations and the formation of protrusions as a result of the impact of shot-peening balls on the machined surface;The vibratory shot-peening process resulted in an increase in Smr (areal material ratio) compared with milling, enhancing the durability and wear resistance of the mating components;The hardened layer thickness g_h_ after the milling process is approximately 30 μm, while the increase in microhardness is below 20 µm;Within the range of parameters used, an increase in amplitude and time results in an approximately linear increase in the ΔHV0.05 coefficient;Amplitude has a greater impact on the increase in surface layer hardening g_h_ compared with time;With an increase in time, the greatest impact on changes in the surface layer hardening g_h_ is observed until t = 13 min, but a further increase in time does not cause significant changes in the g_h_ coefficient;The greatest thickness of the hardened layer, exceeding 200 µm, was obtained after burnishing with an amplitude of A = 60 mm.

## Figures and Tables

**Figure 1 materials-16-06983-f001:**
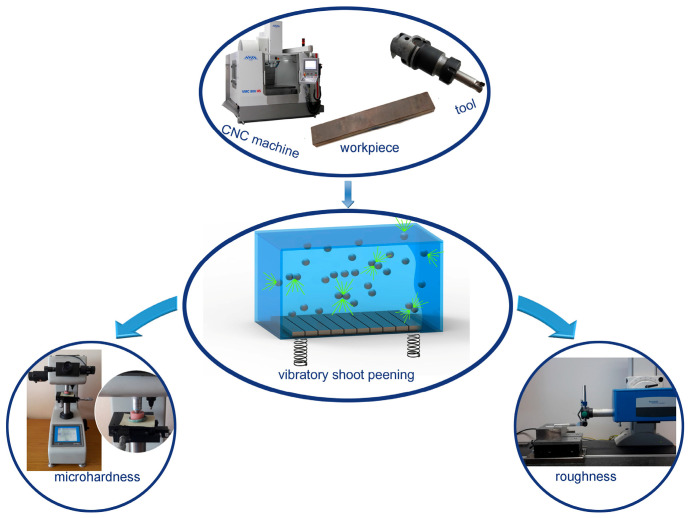
General diagram of the experiment methodology.

**Figure 2 materials-16-06983-f002:**
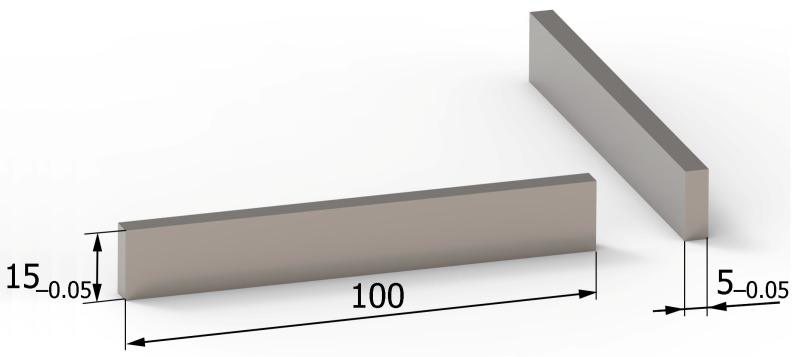
Shape and dimensions (in millimeters) of samples.

**Figure 3 materials-16-06983-f003:**
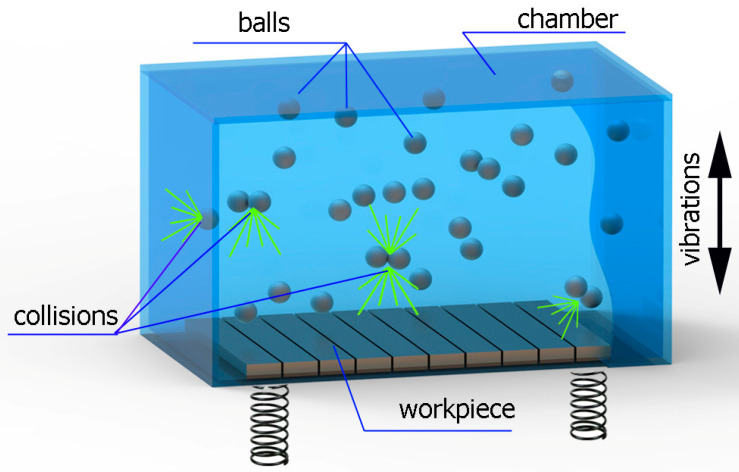
Schematic view of the vibratory shot-peening workstand.

**Figure 4 materials-16-06983-f004:**
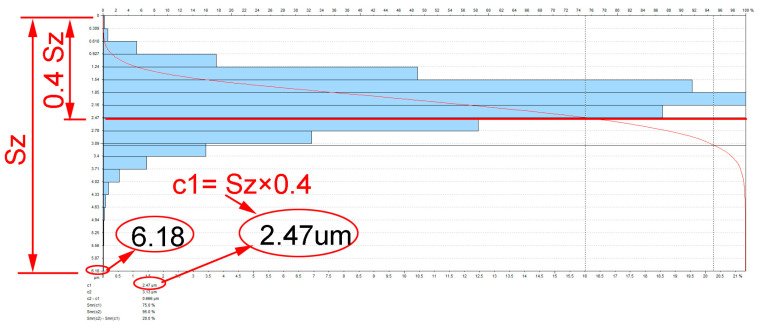
Method of determining height level c.

**Figure 5 materials-16-06983-f005:**
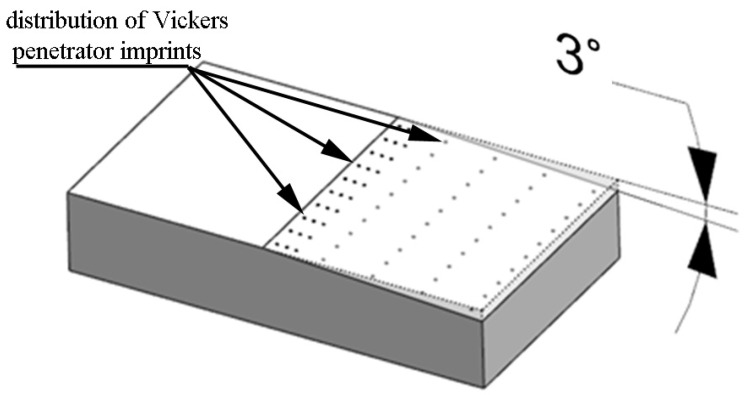
Visualization of distribution of Vickers penetration imprints on oblique sections.

**Figure 7 materials-16-06983-f007:**
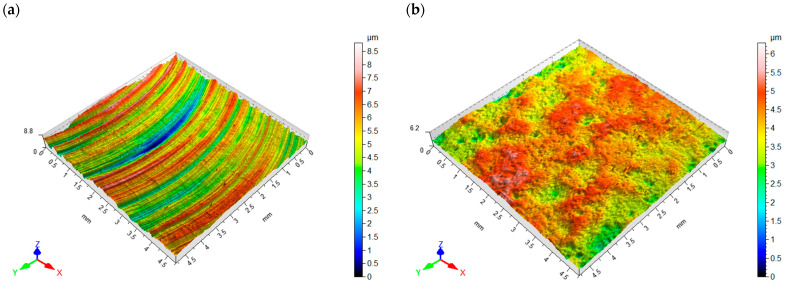
Surface topography: (**a**) after milling, (**b**) after vibratory shot peening.

**Figure 8 materials-16-06983-f008:**
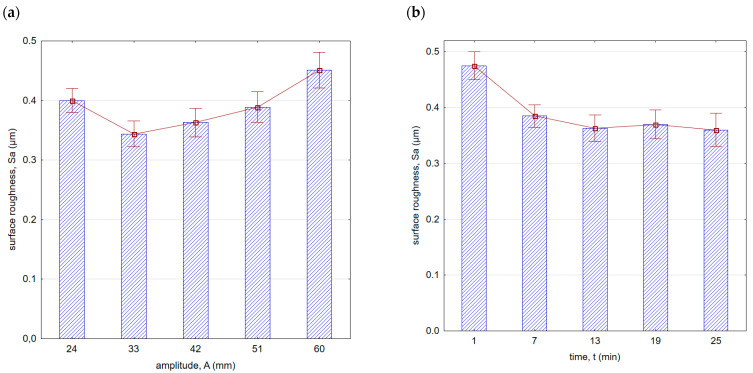
Surface roughness (Sa parameter) after vibratory shot peening: (**a**) variable amplitude (constant t = 13 min), (**b**) variable time (constant A = 42 mm).

**Figure 9 materials-16-06983-f009:**
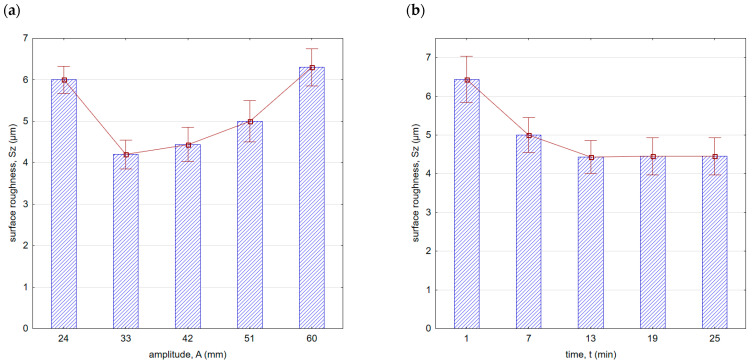
Surface roughness (Sz parameter) after vibratory shot peening: (**a**) variable amplitude (constant t = 13 min), (**b**) variable time (constant A = 42 mm).

**Figure 10 materials-16-06983-f010:**
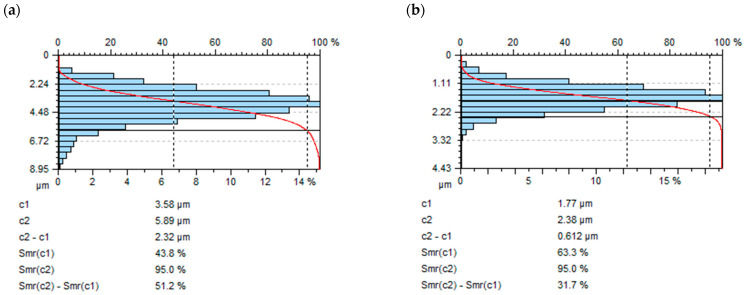
Comparison of areal material ratio curve after (**a**) milling, (**b**) vibratory shot peening (constant A = 42 mm, time t = 13 min).

**Figure 11 materials-16-06983-f011:**
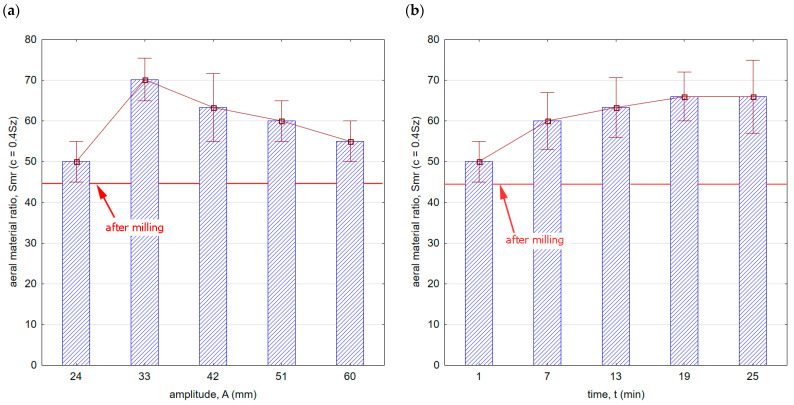
Areal material ratio after vibratory shot peening: (**a**) variable amplitude (constant t = 13 min), (**b**) variable time (constant A = 42 mm).

**Figure 12 materials-16-06983-f012:**
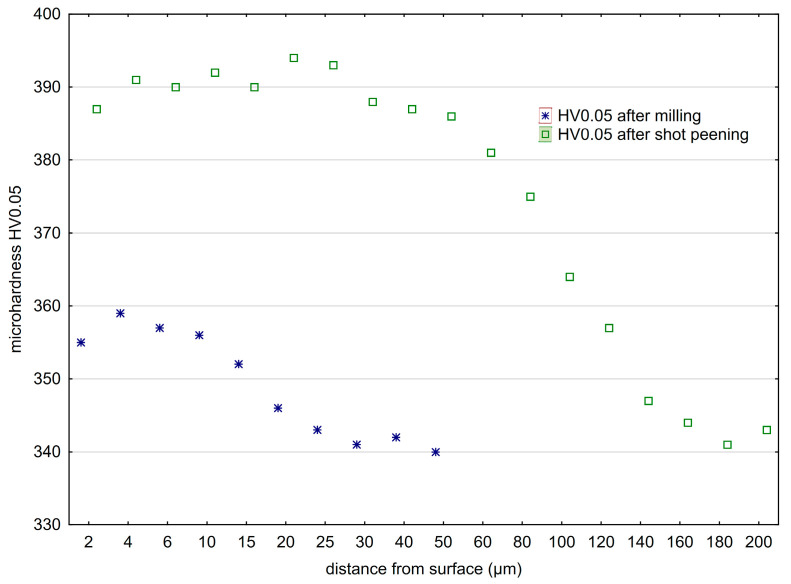
Microhardness distribution of the surface layer of Ti6-Al4-V titanium alloy after milling and shot peening (A = 42 mm, t = 13 min).

**Figure 13 materials-16-06983-f013:**
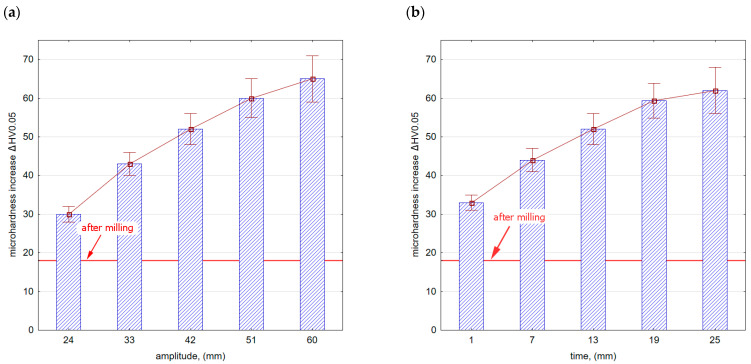
Microhardness increase after vibratory shot peening: (**a**) variable amplitude (constant t = 13 min), (**b**) variable time (constant A = 42 mm).

**Figure 14 materials-16-06983-f014:**
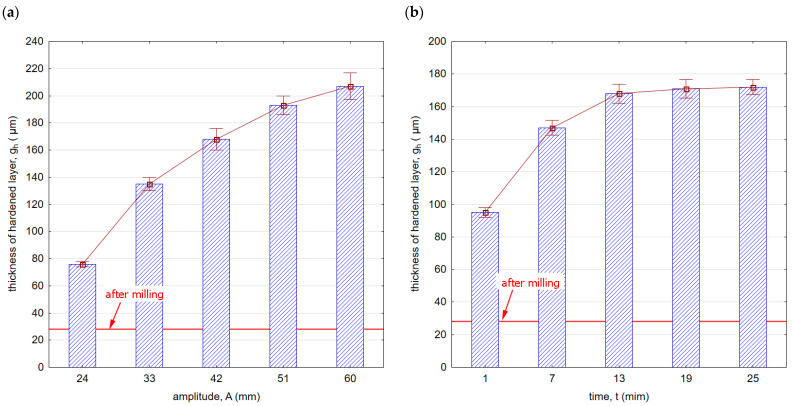
Thickness of hardened layer g_h_ after vibratory shot peening: (**a**) variable amplitude (constant t = 13 min), (**b**) variable time (constant A = 42 mm).

**Table 1 materials-16-06983-t001:** Chemical composition and physical properties of Ti-6Al-4V (Grade 5) titanium alloy.

Chemical Composition, Wt.%		Physical Properties	
Al	6.30	Rm (MPa)	1014
V	4.10		
C	0.026	E (GPa)	120
Fe	0.20		
Ti	Rest	HRC	33

**Table 2 materials-16-06983-t002:** Variable parameters of vibratory shot peening.

No.	Vibration Amplitude A (mm)	Vibration Time t (min)
1	24	13
2	33	
3	42	
4	51	
5	60	
6	42	1
7		7
8		19
9		25

## Data Availability

Not applicable.

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
