# Peer review of "Analysis of Geometric Surface Structure and Surface Layer Microhardness of Ti6Al4V Titanium Alloy after Vibratory Shot Peening"

_materials, 2023, doi:10.3390/ma16216983_

Round 1
Reviewer 1 Report
Comments and Suggestions for Authors
In this paper, the influence of amplitude and time of vibratory shot peening on the surface roughness and physical properties of the surface layer of Ti6Al4V titanium alloy after milling is studied. The results indicate that the surface roughness of the material after vibratory shot peening is significantly reduced compared with milling. In addition, for the increase in hardening of the surface layer, the amplitude has a more remarkable impact compared to the time. It can be considered for publication provided that the following issues are addressed carefully.
1. It is suggested to reorganize the writing logic of the introduction part, especially the last two paragraphs. It is hoped that the key problems to be solved and the main contents involved in this work can be presented more clearly. Rather than just a brief introduction to what a certain literature studies.
2. Since the titanium alloy used is a specific material (Ti6Al4V), its chemical composition needs to be given a specific content, not a range value. In addition, It is suggested to supplement the equipment diagram of vibration shot peening.
3. Why use â–³HV0.05 as an indicator?
4. The conclusion part is recommended to be streamlined, and only important and novel ones need to be listed in the paper.
5. It is suggested that to strengthen the analysis and discussion of experimental results, it should not be just a simple experimental report. That is, The innovation and insight of the paper need to be strengthened.
Comments on the Quality of English LanguageMinor editing of English language required.
Author Response
Article: Analysis of geometric surface structure and surface layer microhardness of Ti6Al4V titanium alloy after vibratory shot peening
Responses to comments from Reviewer 1
I would like to thank the Editor for their consideration, and the Reviewer for the time spent on carefully reviewing this work and for their valuable deep insight and comments. I feel that this paper is now clearer, more thoroughly discussed and better-referenced.
The work has been revised to address the reviewers’ suggestions. Please find hereafter a point-by-point reply to the comments and suggestions. Any revisions to the manuscript was marked up using the “Track Changes”.
General comment from Reviewer 1: In this paper, the influence of amplitude and time of vibratory shot peening on the surface roughness and physical properties of the surface layer of Ti6Al4V titanium alloy after milling is studied. The results indicate that the surface roughness of the material after vibratory shot peening is significantly reduced compared with milling. In addition, for the increase in hardening of the surface layer, the amplitude has a more remarkable impact compared to the time. It can be considered for publication provided that the following issues are addressed carefully
Response:
I would like to thank the Reviewer for his very good opinion. I feel to be obligated to answer for points mention in the review. The paper has been modified and improved. I believe that now is clearer.
Comment 1: It is suggested to reorganize the writing logic of the introduction part, especially the last two paragraphs. It is hoped that the key problems to be solved and the main contents involved in this work can be presented more clearly. Rather than just a brief introduction to what a certain literature studies.
Response:
Thank you for your valuable feedback. I have taken your suggestions into consideration and made the necessary changes to the introduction especially the last paragraph.
Comment 2: Since the titanium alloy used is a specific material (Ti6Al4V), its chemical composition needs to be given a specific content, not a range value. In addition, It is suggested to supplement the equipment diagram of vibration shot peening.
Response:
Thank you for your question. The chemical composition has been given with specific content. Additionally, the vibratory shot peening equipment diagram has been supplemented
Comment 3: Why use â–³HV0.05 as an indicator?
Response:
Due to the fact that the load on the pentrator in the Vickers method was 50g, the microhardness value was expressed in the article as HV0.05. â–³HV0.05 means the increase in relation to the base value (core or annealed samples). The degree of microhardness increase can also be expressed as a percentage, but in this article the â–³HV0.05 indicatior was chosen.
Comment 4: The conclusion part is recommended to be streamlined, and only important and novel ones need to be listed in the paper.
Response:
This is a very important suggestion. I have taken it into account and I removed less important fragments from the "conclusions part", leaving only the most important conclusions
Comment 5: It is suggested that to strengthen the analysis and discussion of experimental results, it should not be just a simple experimental report. That is, The innovation and insight of the paper need to be strengthened.
Response:
I have taken your suggestion into account and expanded the discussion in the article. I have tried more detailed explanations of experimental results.
I appreciate for Editors/Reviewers warm work earnestly, and hope that the corrections will meet with approval. Once again, I thank you very much for your comments and suggestions.
Yours sincerely,
Jakub Matuszak

Reviewer 2 Report
Comments and Suggestions for Authors
major revision

the language of this paper needs more revision
Author Response
Article: Analysis of geometric surface structure and surface layer microhardness of Ti6Al4V titanium alloy after vibratory shot peening
Responses to comments from Reviewer 1
I would like to thank the Editor for their consideration, and the Reviewer for the time spent on carefully reviewing this work and for their valuable deep insight and comments. I feel that this paper is now clearer, more thoroughly discussed and better-referenced.
The work has been revised to address the reviewers’ suggestions. Please find hereafter a point-by-point reply to the comments and suggestions. Any revisions to the manuscript was marked up using the “Track Changes”.
General comment from Reviewer 1: The manuscript entitled “Analysis of the geometric surface structure and surface layer microhardness of Ti6Al4V titanium alloy after vibratory shot penning” discusses the effect of milling and vibratory shot peening on the roughness and microhardness of Ti6Al4V titanium alloy. The review article is made at a good scientific and technical level, and its practical significance is beyond doubt. In order to improve the readability and clarity of the manuscript, some major concerns need to be addressed before the paper is to be accepted for publishing. Detailed comments are as below
Response:
I would like to thank the Reviewer for his very good opinion. I feel to be obligated to answer for points mention in the review. The paper has been modified and improved. I believe that now is clearer.
Comment 1: According to your introduction the Ti6Al4V titanium alloy are largely studied, It must present the originality of your work in the introduction part.
Response:
Thank you for your valuable feedback. The literature review mainly focused on the analysis of the effects after milling of the Ti6Al4V titanium alloy and related burnishing methods. The last paragraph of the literature review additionally highlights the originality of the work and the advantages obtained by using vibratory shot peening after milling the Ti6Al4V titanium alloy, as well as the possibility of using the research results in industry.
Comment 2: Please find in the attachment list of possible references.
https://doi.org/10.1016/j.mtcomm.2021.102428
https://doi.org/10.1016/j.jmrt.2022.06.042
Response:
Thank you for pointing out additional valuable literature references. These references have been included in the literature review.
Comment 3: The abstract part should focus on the scientific findings, and thereby should be deeply revised
Response:
I took your suggestion into account and made the necessary changes to the Abstract. I focused on the most important results of this research work aiming to capture readers' interest and emphasize the significance of our study.
Comment 4: The substrate was not heated during the deposition, but the author claimed that the temperature was kept was 68 °C. How could do this? In my opinion, the temperature might increase with the increase of the deposition time.
Response:
Thank you for your suggestion, the temperature was not measured during the vibratory shot peening process tests, but during the tests no significant increase in the temperature of the shot peened elements was observed as the process time increased
Comment 5: It must to add more detail in the experiment part.
Response:
I have taken your suggestion into account and I have tried more detailed explanations of experiment part.
Comment 6: For the statement: can you rewrite: The surface after vibratory shot peening is characterized by an irregular, non-directional structure obtained as a result of the balls hitting the surface. Mi-cro-irregularity peaks and milling marks are invisible.
Response:
Thank you for the suggestion; striking balls causes the disappearance of traces from milling. The sentence has been rephrased
Comment 7: can you explain the effect of a milling in the surface roughness.
Response:
The same vibratory shot peening conditions can either improve or worsen the surface roughness compared to what it was before shot peening. This is dependent on the roughness after milling. For high values of roughness parameters after milling, the impacting beads 'flatten' the peaks of micro-roughness, reducing the roughness. However, if the initial roughness is very low, the indentations from the balls can lead to an increase in roughness. There is a critical value below which vibratory shot peening cannot go. These conclusions are drawn from other research conducted by me. However, in this article, the influence of the initial roughness (constant milling parameters for all samples) on the roughness after peening was not analyzed.
Comment 8: Can you present the surface topography of the Ti6Al4V titanium alloys after different times and variable amplitude of vibratory shot peening
Response:
Thank you for your valuable attention. The article only shows topographies after milling and one example after vibratory shot peening. Unfortunately, I cannot include the remaining topography due to the use of samples for other research.
Comment 9: “The mechanical impact of the cutting blades on the workpiece material contributes to the plastic deformation of the material in the area of contact between the tool and the workpiece and the increase in discolation in this area, which indirectly affects the hardening of the layer”. It must to add the cress section SEM images to present the change in the morphology by the vibratory shot peening test.
Response:
Thank you for your valuable insight. The information about dislocations was only my observation, not supported by research. However, this analysis was not the subject of this article. In order not to mislead the reader, this fragment has been removed from the manuscript
Comment 10: “Additionally, the heat generated during milling may influence the formation of hard phases”. It must to present the phase formation before and after milling treatment.
Response:
Thank you for your question. The information about phases was only my observation, not supported by research., therefore this fragment was removed from the manuscript
Comment 11: “In turn, as the vibratory shot penning time increases (Fig. 12b), the number of impacts per sample surface, i.e. the degree of coverage of the balls on the surface with impacts, also contributes to the hardening of the surface layer. In both cases, the increase in microhardness significantly exceeds the values obtained after the milling process”. This part is wrong, can you explain the effect of the roughness in the microhardness..
Response:
Thank you for the detailed and thorough analysis of the text. The fragment concerns the influence of vibratory shot peening conditions on microhardness increase ΔHV0.05. The previous version of this passage was not clear and understandable, so it has been replaced as follows: “In turn, as the vibratory shot peening time increases (Fig. 12b), the number of impacts per sample surface increases too, i.e. the degree of ball impact coverage on the surface, also contributes to the hardening of the surface layer. In both cases (influence of amplitude Fig. 12a and time Fig. 12b), the microhardness significantly exceeds the values obtained after the milling process.”
Comment 12: The results and discussion section sort of show and discuss each piece of result one by one. A general discussion is needed to state the vibration mechanism under different times and amplitude and conclude how structure, roughness and morphology affect the hardness. This will strongly strengthen the significance and originality of this manuscript.
Response:
Thank you for your valuable attention. I tried to expand the discussion section and remove insignificant fragments from the conclusions and focus on the most important effects of the research, emphasizing the novelty of the research.
Comment 13: can you present the advantages of this work in the automotive, biomedical and industrial fields.
Response:
thank you for your question. I added a fragment in which I focused on the beneficial effects of vibratory shot peening on the properties of the surface layer and the potential for use in the aerospace and biomedical industries.
Comment 14: The conclusion is very long. The conclusion deserves to be written in such a way that it reflects the importance of the research work.
Response:
This is a very important suggestion. I have taken it into account and I removed less important fragments from the "conclusions part", leaving only the most important conclusions
I appreciate for Editors/Reviewers warm work earnestly, and hope that the corrections will meet with approval. Once again, I thank you very much for your comments and suggestions.
Yours sincerely,
Jakub Matuszak

Round 2
Reviewer 1 Report
Comments and Suggestions for Authors
It is recommended to add the reason for using â–³HV0.05 as an indicator to the text.
Comments on the Quality of English LanguageMinor editing of English language required
Author Response
Responses to comments from Reviewer 1
I would like to thank the Editor for their consideration, and the Reviewer for the time spent on carefully reviewing this work and for their valuable deep insight and comments. I feel that this paper is now clearer, more thoroughly discussed and better-referenced.
The work has been revised to address the reviewers’ suggestions. The English was revised again by the university employee responsible for translations. Please find hereafter a point-by-point reply to the comments and suggestions. Any revisions to the manuscript was marked up using the “Track Changes”.
Comment 1: It is recommended to add the reason for using â–³HV0.05 as an indicator to the text..
Response:
Thank you for your valuable suggestion. information was added to the text and the meaning of the â–³HV0.05 indicator was clarified.
I appreciate for Editors/Reviewers warm work earnestly, and hope that the corrections will meet with approval. Once again, I thank you very much for your comments and suggestions.
Yours sincerely,
Jakub Matuszak
Reviewer 2 Report
Comments and Suggestions for Authors Title: Analysis of the geometric surface structure and surface layer microhardness of Ti6Al4V titanium alloy after vibratory shot peening I accept this paper for publication Comments on the Quality of English LanguageMinor editing of English language required
Author Response
I would like to thank the Editor for their consideration, and the Reviewer for the time spent on carefully reviewing this work and for their valuable deep insight and comments. I feel that this paper is now clearer, more thoroughly discussed and better-referenced.
The work has been revised to address the reviewers’ suggestions. The English was revised again by the university employee responsible for translations. Please find hereafter a point-by-point reply to the comments and suggestions. Any revisions to the manuscript was marked up using the “Track Changes”.
.
I appreciate for Editors/Reviewers warm work earnestly, and hope that the corrections will meet with approval. Once again, I thank you very much for your comments and suggestions.
Yours sincerely,
Jakub Matuszak